# Tocotrienols in Eleven Species of *Hypericum* Genus Leaves

**DOI:** 10.3390/molecules30030662

**Published:** 2025-02-02

**Authors:** Danija Lazdiņa, Inga Mišina, Paweł Górnaś

**Affiliations:** Institute of Horticulture, Graudu 1, LV-3701 Dobele, Latvia; danija.lazdina@llu.lv (D.L.); inga.misina@llu.lv (I.M.)

**Keywords:** Guttiferaceae, St. John’s wort, herb, lipophilic bioactive compound, phytochemical, antioxidant, vitamin E, TOCOL

## Abstract

Saint John’s worts or goatweeds are mostly perennial flowering plants in the *Hypericaceae* family, formerly under the *Clusiaceae* family. Teas and macerations of the plants are common in traditional medicines and modern depression and cancer therapies. The most notable bioactive compounds in *Hypericum* are hyperforin and hypericin. While *Hypericum* contains a variety of carotenoid and phenolic compounds, which are well documented, there is little available information on tocopherols and almost none on tocotrienols. Considering the frequency of tocotrienol derivatives in *Clusiaceae* species, this study investigates and reports the presence of tocotrienols in eleven *Hypericum* species’ leaves: *H. hircinum*, *H. hookerianum*, *H. calycinum*, *H. xylosteifolium*, *H. densifolium*, *H. prolificum*, *H. kalmianum*, *H. frondosum*, *H. olympicum,* and two hybrids: *H.* × *moserianum* and *H* × ‘Rowallane’. Eight tocopherol and tocotrienol forms (α, β, γ, δ) were detected in the leaves, predominantly containing α-tocopherol. Tocotrienol content was most significant in *Myriandra* section species and was highest in *H. prolificum* (22.90 ± 0.63 mg 100 g^−1^), while the highest tocotrienol proportion was observed in *H.* × ‘Rowallane’ (54.12% of total tocochromanols) and *H. prolificum* (37.27% of total tocochromanols). The results demonstrated significant tocochromanol accumulation in *Hypericum* leaves.

## 1. Introduction

*Hypericum* is a genus of flowering mostly perennial plants in the Hypericaceae family. Various species are common across the globe and are used in phytomedicines [1,2] or as decorative plants. *Hypericum* plants contain a variety of bioactive compounds, but the most notable are hyperforin and hypericin. Hyperforin is a prenylated phloroglucinol, which acts as a mild antidepressant [3,4,5,6]. Hypericin is a polycyclic anthraquinone with strong photo-sensitizing properties used in photo-dynamic cancer treatment [7,8] and has antiviral properties against human immunodeficient virus [9,10], hepatitis C [11], and other enveloped and non-enveloped viruses [12,13,14].

Hypericin, hyperforin, and related compounds are the most concentrated in the reproductive organs of the plant during full flowering, while the content in leaves is generally lower [15,16] even though synthesis is performed in the leaves as well. Hypericin and hyperforin are produced and accumulated in dark and lighter translucent glands, respectively, and are not found in non-secretory tissue [17]. While dark glands, in which hypericin is accumulated, are located on all parts of the plant, translucent glands are especially concentrated in leaves [18]. It is hypothesized that hyperforin shares a non-mevalonate (methylerythritol phosphate, MEP) biosynthetic pathway (geranyl pyrophosphate) with monoterpenes in gland cell chloroplasts on account of a shared biosynthetic pathway with essential oil constituents present in translucent glands [19,20]. The lower content of *Hypericum*-relevant phytochemicals in the leaves of the most well-studied species has caused *Hypericum* leaves to be understudied.

Tocochromanols are a class of lipophilic antioxidants. They consist of a methylated homogentisate-derived chromane ring and a lipophilic tail. The methyl group position dictates the tocochromanol homologue (δ-, β-, γ-, α-tocochromanol), while the tail determines tocochromanol species (tocopherol/tocotrienol/plastochromanol-8). The most common representatives are tocopherols (Ts), tocotrienols (T3s), and plastochromanol-8. Tocopherols are the most common and well studied. They are present in all plant tissue, usually in the form of α-T or γ-T, while other tocochromanols are more common in plant seeds [21]. While homogentisate is used in the biosynthesis of all tocochromanols, each uses a different precursor compound in the lipophilic tail. Tocopherols use phytyl pyrophosphate (PPP), and tocotrienols use geranylgeranyl pyrophosphate (GGPP). The PPP used to synthesize tocopherols can be derived from either the MEP pathway or degraded chlorophyll through phytol recycling [22], while GGPP is exclusively MEP pathway-derived. Tocochromanol biosynthesis converges to a common biosynthetic pathway once the lipophilic tail is added to the homogentisate by a respective homogentisate transferase, with the common enzymes being methyl-transferase (adds methyl group that is present γ-, α-tocochromanols and plastochromanol-8), tocopherol cyclase (cyclizes the homogentisate ring to produce a chromane ring) and γ-tocopherol methyltransferase (adds methyl group to δ- and γ-tocochromanols to produce β- or α-tocochromanols, respectively) [23].

The most common use of Ts and T3s is as natural lipophilic antioxidants in food and cosmetics, but numerous studies have demonstrated their potential in the prevention and treatment of cardiovascular and neurodegenerative diseases, conditions related to metabolic syndrome, oxidative stress, anti-tumoral activity [24].

Like hypericin and hyperforin, tocochromanols are produced in cell chloroplasts, and anthraquinones (hypericin) share biosynthetic pathways with tocochromanols, as the shikimate, MEP and mevalonate (MVA) pathways are hypothesized in both [25,26]. The exact biosynthesis route of anthraquinones in plants is not known. In *Senna tora*, the primary initial precursor is malonyl-CoA, which is successively condensed into a linear polyketide chain and then cyclized and decarboxylated to produce anthrone units such as emodin anthrone [27]. Hypericin is synthesized through the oxidative dimerization of emodin anthrone and protohypericin [28]. The phloroglucinol moiety in hyperforin is synthesized via the polyketide mechanism, while the isoprenoid moieties in hyperforin are derived from the MEP pathway [18,20].

Genus-specific anthraquinones, phloroglucinols and tocotrienols are not unique to *Hypericum*, but are relatively widely reported in other former and current Clusiaceae family members, although detailed profiles are available only for certain plant parts. While other members in the Malpighiales order, to which the Clusiaceae, Hypericaceae, and Calophyllaceae belong, are less studied, the consistent presence of phloroglucinols and tocotrienols in current and former Clusiaceae members implies some relationship between the synthesis of these compounds, especially considering that the pathways and genetic regulation of anthraquinone and prenylated phloroglucinol biosynthesis are not yet deciphered and detailed descriptions exist for *Senna tora* (Fabaceae, Fabales order) and *Rheum tanguticum* (Polygonaceae, Caryophyllales order).

While *H. perforatum* tocochromanol content has been studied, this study did not use tocotrienol standards [29]. Another investigation of tocopherols and tocotrienols used a mass spectrometry-based assay and noted both α-T and δ-T3 in *H. perforatum* leaves [30]. As a result, the presence of tocotrienols in the *Hypericum* genus has gone mostly unreported. This study reports on the presence of tocochromanols (tocopherols and tocotrienols) in *Hypericum* species’ leaves and presents them as a source of tocotrienols. The presence of tocotrienols in *Hypericum* leaves implies expanded use of the plants in the pharmaceutical industry and biochemical studies, considering tocotrienols are scarcely reported in photosynthetic plant organs and are believed to only accumulate in the seeds. Such an approach would contribute to a more efficient use of the harvested plant material *Hypericum* genus.

## 2. Results and Discussion

The leaves of eleven *Hypericum* species belonging to five sections were tested: *Androsaemum* section (*H. hircinum*), *Ascyreia* (*H. hookerianum*, *H. calycinum*), *Inodora* (*H. xylosteifolium*), *Myriandra* (*H. densifolium*, *H. prolificum*, *H. kalmianum*, *H. frondosum*), and *Olympia* (*H. olympicum*), as well as two hybrids: *H. × moserianum* (hybrid of *H. calycinum* and *H. patulum*), *H.* × ‘Rowallane’ (hybrid of *H. leschenaultii* and *H. hookerianum*), both categorized under *Ascyreia* based on parent species. Figure 1 illustrates the RP-HPLC-FLD chromatograms of the tocochromanols profile in the leaves of *H. hookerianum*, *H. calycinum*, and the standard solution.

The profile of tocochromanols in different *Hypericum* species varied considerably. Along with the eight known tocopherols and tocotrienols *Hypericum* species, several unidentified compounds formed peaks in the chromatograms. However, these could not be identified from the elution time and UV spectrum alone (Figure 1 and Appendix A) and would require identification with mass spectrometry, which could not be performed during this study. Reverse-phase (RP) high-performance liquid chromatography (RP-HPLC) with a fluorescent light detector (FLD) is currently one of the most popular techniques used to determine tocochromanols [31]. Provided effective peak separation, the FLD and diode array detector (DAD) combination is generally sufficient to confirm tocohromanol identity, but additional confirmation with mass spectrometry is advisable if only trace amounts of the compound are present, or the material is likely to contain unique or uncommon compounds [32]. Considering the presence of tocotrienols and unidentified phytochemicals in the *Hypericum* genus and related genera, verification with MS is advisable in future studies. The potential for co-elution of other compounds with tocopherols and tocotrienols is also likely and should be considered.

The results of tocopherol and tocotrienol concentration are presented in Appendix A and Figure 2.

The tocochromanol profile was dominated by α-T (34.81–95.63% of total tocochromanols (TT)) in most of the tested samples, usually followed by γ-T3 (up to 33.79% in *H*. × ‘Rowallane’) or α-T3 (up to 32.72% in *H. prolificum*). Additionally, *H. prolificum* had a much higher γ-T content and proportion than other samples. Of the two analyzed hybrids, *H.* × *moserianum* and *H.* × ‘Rowallane’, the first had a proportionally similar tocochromanol profile to *H. calycinum*, one of its parent species, while *H.* × ‘Rowallane’ total tocochromanol content was lower, especially α-T, and the tocotrienol proportion was much higher, especially γ-T3. Hybrid *Elaies guineensis* × *E. oleifera* oil palms have shown different tocotrienol/tocopherol proportions; however, the content and proportion are within the range for non-hybrid *E. guineensis* and *E. oleifera* [33,34].

The results agree with a 2012 study on tocochromanol composition in *H. perforatum*, where α-T constituted a majority of the leaf tocochromanols, preceded by a much smaller δ-T3 in the chromatogram and some small, but unidentified peaks in between [30]. In 2017, a report on lipid contents in *H. perforatum*, *H. perfoliatum, H. tomentosum*, and *H. ericoides* was published and presented δ-T as the dominant tocochromanol; however, this study did not use tocotrienol standards [29]. Tocopherol contents have not previously been investigated in any of the species included in the present study. The one study that has investigated tocopherols in the top two-thirds of *H. perforatum*, *H. perfoliatum*, *H. tomentosum,* and *H. ericoides* observed δ-T at the highest concentration, while γ-T and α-T content was much lower in all four species. This study did not observe β-T in the samples and did not investigate tocotrienols. The total tocochromanol content was much lower (3–3.5 mg 100 g^−1^) [29] than observed in the present study (the lowest total tocopherol content was 17.35 mg 100 g^−1^ in the ‘Rowallane’ hybrid). It is, however, highly unusual for leaf tocochromanols to be dominated by δ-T in any species regardless of stress or growth stage [22,35,36]. Although a different organ and different species were investigated in the present study, the variation between species in the present study is relatively low within either study. Tocochromanol and tocopherol, especially α-T, content can be higher in mature leaves than in other plant organs [37], and much of the difference can be explained by differences in sampling and plant organs used for tocochromanol extraction.

More recently, a study investigated tocochromanol extraction from *H. perforatum* inflorescences using hydroethanolic solutions and compared it with the saponification protocol used in the present study. All common tocopherol and tocotrienol homologues were observed and the tocotrienol proportion was higher. δ homologues were more effectively recovered in hydroethanolic solutions than α homologues [38]. A major difference between the present and two early studies is the sample preparation protocol—methanolic extracts were used in 2012 and hexane extracts were used in 2017 with no saponification, while the present study tocochromanols were extracted in ethanol, saponified, and re-extracted in a mixture of *n*-hexane–ethyl acetate (9:1, *v*/*v*). Extraction in different solvents can cause drastically different solubility of individual tocochromanols and different tocochromanol profiles [39]. Similarly, saponification versus extracting directly in ethanol affects tocochromanol composition [40]. Saponification allows for superior extractability of both tocopherols and tocotrienols from plant matrices—it damages cell walls, allowing for better solvent access, and releases esterified tocochromanols, which are the most widely reported type of tocochromanol derivative [31]. Lower extractability can be caused by the presence of bound tocochromanols such as tocopheryl fatty acid esters in plant material [31,41,42] or non-extractable tocochromanols [31,43]. Of the observed tocochromanols, δ-T3 required the lowest concentration of ethanol in direct extraction protocols, and all tocochromanol recovery increased with a higher ethanol proportion in the solvent [38]. The solubility of tocochromanols in hydroethanolic solutions is influenced by the degree of saturation, the length of the side chain of the molecule, and the number of methyl substituents on the chromanol ring [38,43]. Efficient tocochromanol extraction (tocopherol and tocotrienol) does not appear to require the most hydrophobic solvent but a particular polarity [39,40]. Of the mentioned solvents, hexane (0.009) and ethyl acetate (0.228) have the lowest relative polarity. Acetone (0.355) has a low polarity, while 2-propanol (0.546), ethanol (0.654), and methanol (0.762) are increasingly more polar. Despite the difference in polarity, extraction directly in ethanol has yielded similar results to saponification in ethanol and subsequent re-extraction in hexane–ethyl acetate mixtures.

While differences to data provided in existing studies are easily explained with differences in sample preparation, the differences between different species tocochromanol profiles are not. Full tocochromanol profiles are seldom explored in leaf samples across several species within a genus. Rosaceae fruit tree leaves exhibited a similar degree of variability between species when harvested at the same time from trees growing in similar conditions [35]. Seed tocochromanol profiles are strongly phylogeny shaped in the Fabaceae family [44,45], which is generally tocopherol-dominated, and differ significantly between different Apiaceae species’ seeds as well, though formal studies on tocochromanol relation to phylogeny have not been published [46]. The T, T3, and TT contents observed in the present study were higher than previously reported in other leaves and green vegetables. For reference, broccoli contains 3.75 mg 100 g^−1^ [47] up to 22.1 mg 100 g^−1^ of α-T [48] and spinach contains 1.87–5.9 mg 100 g^−1^ of α-T and 2.78–4.18 mg 100 g^−1^ of γ-T [49,50], while T3s are generally either absent or present in only very small concentrations. Relatively high α-T concentration has been observed in moringa (18.3 mg 100 g^−1^), cassava (7.1 mg 100 g^−1^) leaves [51], and nettle (*Urtica leptophylla*) leaves (11.93 mg 100 g^−1^) from Costa Rica [37].

Several, but not all, of the investigated species had high T3 content and proportion. While the presence of T3s in photosynthetic tissue is not unprecedented [37], it is uncommon. Tocotrienol derivatives have previously been found in several *Clusia* [52,53,54] and *Garcinia* [55,56] species’ stem bark and leaves in addition to T3s in *Calophyllum inophyllum* seeds [21], which used to be classified under the same family as *Hypericum*; T3s are present in other families in the Malpighiales order, such as *Hevea brasiliensis* latex in the Euphorbiaceae family [21]. While the common presence of δ-T3 derivatives in closely related families may raise some suspicion about the verity of the results in the present study, the previously mass spectrometry-confirmed presence of δ-T3 in *H. perforatum* leaves confirms the results [30]. In the current study, a saponification process was used, which is the most effective protocol for the recovery of tocochromanols from plant material [31], but does not account for tocochromanol acids, derivatives, and dimers due to the alkaline reaction.

According to principal component analysis (PCA), principal components (PCs), PC1 and PC2, explain 73 and 20.1% of the total variance, respectively, for a total of 93.1% explained variance. PC1 had high loadings with α-T (0.98) and γ-T3 (−0.12), while PC2 had high loadings with α-T3 (−0.81), γ-T (−0.56) and γ-T3 (0.14); other variable loadings for the two components were minor (−0.09 to 0.09). In total, α-T had the highest contribution between the two PC loadings. As shown in Figure 3, there is some relation to the taxonomic classification of the plants—*Ascyreia*, *Inodora*, and *Androsaemum* members tend to have leaves richer in γ-T3 along *H. kalmianum*. These form a loose, extended cluster. Another is formed by *H. frondosum*, *H. densiflorum* (*Myriandra* section), and *H. olympicum* (*Olympia* section), with *H. prolificum* distant from either cluster. This relates back to the proportionally similar tocochromanol profile in the *Ascyreia* section and the diverse proportional tocochromanol profile in the *Myriandra* section. The two hybrid species included in this study, *H.* × ‘Rowallane’ and *H.* × *moserianum*, were located at the extreme PC1 coordinates of the *Ascyreia* section cluster. As evidenced by PCA, α-T, α-T3, γ-T, and γ-T3 are the main delineators between *Hypericum* leaf tocochromanol composition according to species used in the present study.

The tocopherol concentrations observed in the present study are similar to vegetable oils. Most of the tested samples have similar α-T content to sunflower (32.7–59.0 mg 100 g^−1^), safflower (36.7–47.7 mg 100 g^−1^), and cottonseed (30.5–57.3 mg 100 g^−1^) oils. The content of α-T3 and γ-T3, which were the main leaf tocotrienols, is not significantly higher than some oilseeds, such as grapeseed oil, which can contain 15.2–21.6 mg 100 g^−1^ α-T3 and 25–32 mg 100 g^−1^ γ-T3 or 36.87–46.33% (γ-T3) and 22.42–28.88% (α-T3) of total tocochromanols [57]. The α-T3 content in *H. prolificum* (20.1 ± 0.55 mg 100 g^−1^) is within the upper range observed for palm (5.7–26.0 mg 100 g^−1^) oil [24]. The annual production, oil content, tocochromanol content, and tocotrienol proportion in *Hypericum* leaves are lower than grape seeds, making them a more expensive source of tocotrienols.

The characteristic phytochemicals present in *Hypericum* plants, including the leaves, are extractable in aqua-alcoholic solutions, pure ethanol [58], supercritical CO_2_ [59,60], and, to a limited degree, oils [61]—they are co-extractable with tocochromanols but were not analyzed in the present study. Relationships between hypericin, hyperforin, and tocochromanol biosynthesis cannot be surmised from the produced data, and there are no studies on the biosynthetic relationship between these compounds or their groups, and very little investigation has been done on tocotrienol content in *Hypericum* leaves.

A δ-tocotrienoloic acid derivative has been isolated from *Clusia pernambucensis* stem bark [62], *C. melchiorii* trunk [53], 5-hydroxy tocotrienoloic acid (garcinoic acid), and 5-hydroxy-8-methyltocotrienol along with other tocotrienol-related compounds from *C. minor* leaves, flower buds, and blossomed flowers [63], and polyprenylated phloroglucinol 6S,8S,28S-nemorosic acid has been observed in *C. nemorosa* fruits [64]. Tocotrienols and their derivatives are widely reported in the *Garcinia* genus—the stem bark of *G. amplexicaulis* contains unique tocotrienol derivatives, δ- and γ-amplexichromanol, along with their derivatives and common tocochromanols [65,66]; the fruits of *G. paucinervis* contain polyprenylated acylphloroglucinols (paucichymols), a δ-tocotrienol derivative [55], and other tocotrienol derivatives, garcipaucinones [67]; the fruits of *G. multiflora* contain garcimultinones, a variety of homoadamantane polycyclic phloroglucinols [68,69]; the fruits of *G. oblongifolia* contain tocotrienols and their derivatives, including a dimeric tocotrienol derivative and several prenylated phloroglucinols [70], while the leaves contain garcinoic acid and a variety of benzophenones, polyprenylated acylphloroglucinols common in the genus [71]; the stem bark of *G. virgata* contains δ-tocotrienol formylated tocotrienols and cotoin (benzophone derivative) [72]; the leaves of *G. nigrolineata* contain a tocotrienol quinone dimer [56]; and *G. kola* seeds contain garcinoic acid [73]. Another former member of the Clusiaceae family, the *Calophyllum* genus, also produces and accumulates tocotrienols, as δ-T3 has been observed in *Calophyllum thorelii* bark [74], and *C. calaba* and *C. inophyllum* kernel oils have significant tocotrienol content [75].

The common presence of tocotrienols and their derivatives along with phloroglucinols is not explained by a common synthetic pathway. Meanwhile, anthraquinones, which do have a common biosynthetic pathway with tocochromanols, are only reported in *Hypericum* species. Moreover, tocotrienol derivatives, but not tocotrienols, are reported in *Garcinia* and *Clusia* species, while the present study found non-derivatized tocochromanols. Additional metabolic and genomic studies are necessary to elucidate possible connections between tocotrienol, phloroglucinol, anthraquinone biosynthesis, regulation, and relation to plant taxonomy, as little information is available on Clusiaceae genera, except *Clusia* and *Garcinia*, and other species formerly categorized under the Clusiaceae family.

## 3. Materials and Methods

### 3.1. Reagents

Ethanol, methanol, ethyl acetate, *n*-hexane (HPLC grade), pyrogallol, sodium chloride, and potassium hydroxide (reagent grade) were purchased from Sigma-Aldrich (Steinheim, Germany). Ethanol (96.2%) for leaf sample saponification was received from SIA Kalsnavas Elevators (Jaunkalsnava, Latvia). Standards of tocopherol homologues (α, β, γ, and δ) (>98%, HPLC) were obtained from Extrasynthese (Genay, France), while tocotrienol homologues (α, β, γ, and δ) (>98%, HPLC) were from Cayman Chemical (Ann Arbor, MI, USA).

### 3.2. Plant Material

The softwood cuttings of eleven *Hypericum* species, *H. hircinum* (item no. S-71693), *H. hookerianum* (item no. S-879), *H. calycinum* (item no. w), *H. xylosteifolium* (item no. S-63126)*, H. densifolium* (item no. S-74255), *H. prolificum* (item no. S-44847), *H. kalmianum* (item no. S-834), *H. frondosum* (item no. S-49853), and *H. olympicum* (item no. n), as well as two hybrids: *H. × moserianum* (item no. S-22666) and *H.* × ‘Rowallane’ (item no. C-444), were cut 5–20 cm from the branches of healthy plants of *Hypericum* in morning hours (8:30–10:30) on 8 July 2022 in the National Botanic Garden of Latvia (NBG) in Salaspils. The present study investigated all *Hypericum* species and varieties grown in NBG, including two hybrids, for the most comprehensive data pool possible. It was optimal time (shoot maturing stage) to obtain cuttings, which were originally intended for plant propagation. From each *Hypericum* species, cuttings were collected from 1 to 5 different plants, resulting in 3–10 cuttings each. The cuttings were packed in sealed bags and sprayed with water to keep plants alive during transport to the lab in the Institute of Horticulture, Dobele, Latvia. On the same day after transport, softwood cuttings were prepared for propagation by external acting and removal of most of the leaves. To obtain represented biological replication of leaves for each *Hypericum* species, leaves were separated from different grouped softwood cuttings. Each biological sample of leaves was frozen at −80 °C separately, stored at this temperature for a week, and then freeze–dried using a FreeZone freeze–dry system (Labconco, Kansas City, MO, USA) at a temperature of −51 ± 1 °C in a vacuum (pressure below 0.01 mbar) for 48 h. The dried plant material for each sample (1–10 g) was completely powdered using an MM 400 mixer mill (Retsch, Haan, Germany). The parameters during milling were as follows: a frequency of 30 Hz within seconds and time of 60 s. The obtained 5 μm final fineness (according to the manufacturer) powder was used directly for tocopherol and tocotrienol homologue extraction as described below in Section 3.3. Dry mass was measured gravimetrically. The leaves were processed and analyzed within a month of cutting collection (from receiving plant material to HPLC analysis).

### 3.3. Saponification and n-Hexane–Ethyl Acetate Extraction Protocol

Saponification protocol was performed as described earlier [76], with small modifications—the absolute ethanol was replaced by 96.2% ethanol to lower costs. Changes from absolute to 96.2% ethanol did not affect statistically significantly (*p* < 0.05) results (Appendix A). An amount of 0.1 g powdered leaf sample was placed in a 15 mL glass tube with screw cap. Then, 0.05 g of pyrogallol was added to prevent oxidation of tocopherols and tocotrienols. The mixture was sequentially supplemented with 2.5 mL of 96.2% ethanol and mixed. The process of saponification was incited by adding 0.25 mL of 60% (*w*/*v*) aqueous potassium hydroxide. The glass tube was immediately closed with a screw cap, mixed for 10–15 s using vortex REAX top (Heidolph, Schwabach Germany) with vibration frequency rates up to 2500 rpm, and sequentially subjected to incubation in a water bath at 80 °C. After 10 min of incubation, sample was mixed again for 10–15 s using vortex REAX top at 2500 rpm. After 25 min of incubation to stop/slow down the process of saponification, sample was cooled immediately in an ice-water bath for 10 min. The process of tocopherol and tocotrienol homologues extraction started by adding 2.5 mL of 1% (*w*/*v*) sodium chloride to the glass tube with the sample to lower the surface tension between the two non-miscible solvents (hydro ethanol and *n*-hexane–ethyl acetate), which were mixed for 5 s using vortex REAX top at 2500 rpm. Then, 2.5 mL of *n*-hexane–ethyl acetate (9:1; *v*/*v*) was added to extract the tocopherol and tocotrienol homologues and mixed for 15 s using vortex REAX top at 2500 rpm. After mixing with the organic solvent mixture (*n*-hexane and ethyl acetate), sample was centrifuged for 5 min (1000× *g*, at 4 °C). The organic layer, containing *n*-hexane and ethyl acetate, was moved to a 100 mL round bottom flask. The extraction residues were re-extracted in a fresh portion of 2.5 mL of *n*-hexane–ethyl acetate (9:1; *v*/*v*) as described above. Re-extraction was performed two times. The organic layer from initial extraction and two re-extractions was collected and combined in the same 100 mL round bottom flask and evaporated in a vacuum rotary evaporator Laborota 4000 (Heidolph, Schwabach, Germany) at 40 °C until fully dry. The obtained thin film layer on the bottom of the flask was dissolved in 1 mL ethanol (HPLC grade) and transferred to 2 mL analytical glass vial.

### 3.4. Tocopherol and Tocotrienol Determination by RP-HPLC-FLD

Tocopherol and tocotrienol homologues were determined according to the previously developed and validated method [77]. Tocochromanols analysis was performed using reverse-phase high-performance liquid chromatography with fluorescent light detector (RP-HPLC-FLD) via HPLC system (Shimadzu Corporation, Kyoto, Japan) consisting of a pump (LC-10ADvp), a degasser (DGU-14A), a low-pressure gradient unit (FCV-10ALvp), a system controller (SCL-10Avp), an auto-injector (SIL-10AF), a column oven (CTO-10ASvp), and a fluorescence detector (RF-10AXL). The chromatographic separation of tocopherol and tocotrienol homologues was carried out on the Luna PFP(2) (pentafluorophenyl phase) column with the following parameters: particle morphology—fully porous; particle size—3 µm; column length—150 mm; and column ID—4.6 mm. It was secured with a guard column of the length—4 mm and ID—3 mm (Phenomenex, Torrance, CA, USA). The chromatography analysis was performed under the isocratic conditions as follows: mobile phase—methanol with water (93:7; *v*/*v*); flow rate—1.0 mL/min; column oven temperature—40 °C; room temperature—22 ± 1 °C. The total chromatography runtime was 13 min. The identification and quantification were performed using a fluorescence detector at an excitation wavelength of 295 nm and emission wavelength of 330 nm. The quantification was performed based on the calibration curves obtained from tocopherol and tocotrienol standards.

### 3.5. Statistical Analysis

The results were presented as means ± standard deviation (*n* = 3) from three independent replications (biological samples) of the plant material. The means ± standard deviations were calculated in Microsoft Excel (Version 1808, Microsoft Office, Redmond, WA, USA). The PCA was performed using open-source R libraries psych and factoextra in RStudio 2024.12.0+467 “Kousa Dogwood” Release (cf37a3e5488c937207f992226d255be71f5e3f41, 2024-12-11) (Posit Software, Pbc, Boston, MA, USA) without data scaling to avoid noise from minor compounds using species’ mean tocochromanol contents. Visualizations were produced using MS Excel and R open-source libraries ggplot2, ggthemes, and ggrepel.

## 4. Conclusions

While, like most plants, *Hypericum* leaf tocochromanols are mainly composed of α-T (35–96%, averaging 72% of TT), a significant portion of tocochromanols are made up of tocotrienols (2–54%, averaging 20% of TT), most notably α-T3 and γ-T3. This discovery is unique due to the rarity of tocotrienol observations in photosynthetic tissues, as reported by existing articles. There is some relationship between the section and tocochromanol composition—*Androsaemum* and *Inodora* plant leaves contained only trace amounts of tocotrienols, while tocotrienols can make up a majority of leaf tocochromanols in *Ascyreia* and *Myriandra* species’ leaves. The differences between tocochromanol composition in different sections imply some connection to plant phylogeny, but the tocochromanol profile is not solely dictated by it. Considering the high tocotrienol contents in *Hypericum* species’ leaves, further studies into *Hypericum* and related plant tocotrienol contents are warranted. Investigating tocotrienols, as opposed to just tocopherols or just α-T, is advisable for other plant materials as well. Most existing reports on *Hypericum* phytochemicals and their antioxidant, bioactive, and medicinal properties have not addressed the potential presence of tocotrienols, which possess antioxidant, bioactive and medicinal properties. In fact, most studies of tocochromanols in plant leaves only investigate tocopherols, regardless of the plant family or tocotrienol’s presence in non-photosynthetic tissue. Investigating tocotrienol contents in photosynthetic tissues of other plant families may reveal unreported sources of tocotrienols and provide information on plant tocochromanol metabolism.

Given the complex profile of tocochromanols in the leaves of the *Hypericum* genus, it is advisable to explore a broader range of species and tocochromanols and perform their identity verification with mass spectrometry (MS) and/or nuclear magnetic resonance (NMR) tools.

Considering the significant amounts of tocotrienols detected in *Hypericum* leaves, these ornamental plants could be used to extract these rare lipophilic phytochemicals for pharmaceutical purposes. Additional focus on the co-extraction of tocotrienol, their compounding effects, and medicinal properties is warranted in studies on hyperforin and hypericin extracts since the co-extraction of tocotrienols is well documented under common hypericin and hyperforin extraction conditions in alcohol and hydro-alcoholic solutions. Moreover, additional studies on tocotrienol separation and purifications are warranted.

## Figures and Tables

**Figure 1 molecules-30-00662-f001:**
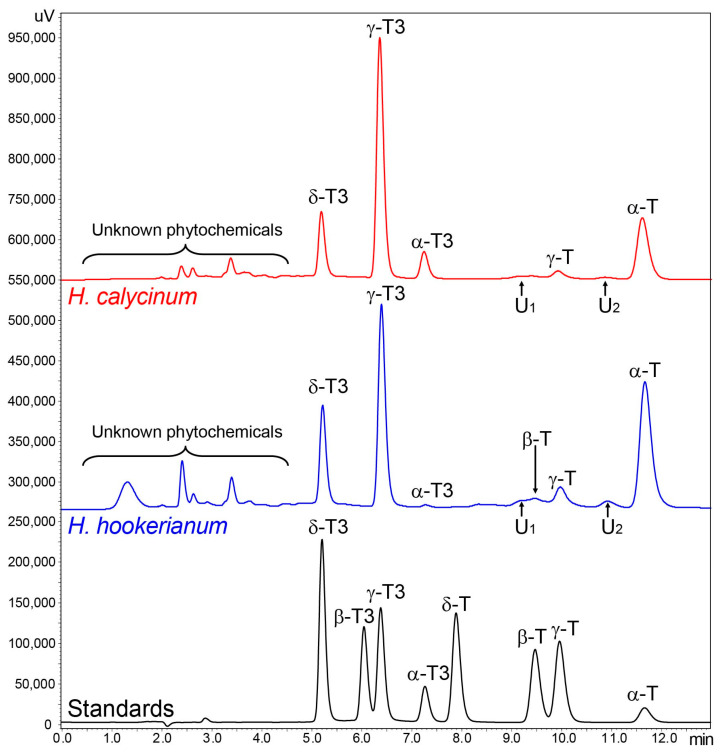
Chromatograms of the tocotrienol (T3) and tocopherol (T) homologues (α, β, γ, and δ) separation by RP-HPLC-FLD in the leaves of *H. hookerianum*, *H. calycinum,* and standards. U: unknown compound.

**Figure 2 molecules-30-00662-f002:**
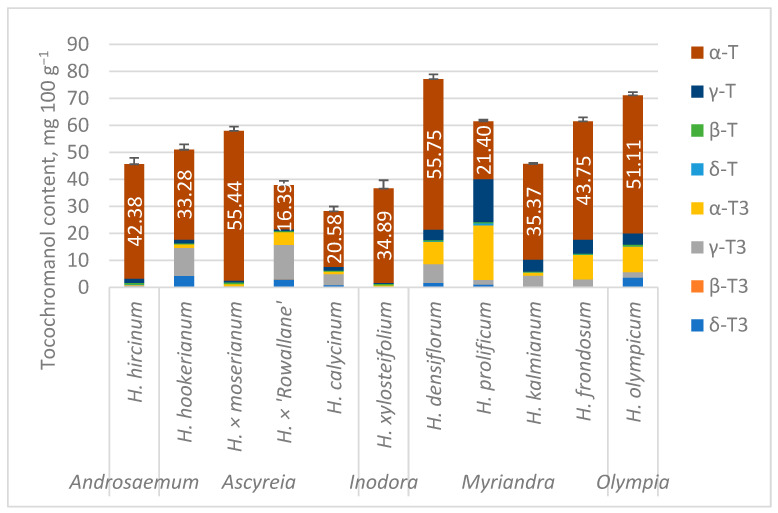
Tocochromanol contents in *Hypericum* leaves. Data are presented as stacked mean content of three replications + standard deviation of total tocochromanol content, and values present mean α-T content.

**Figure 3 molecules-30-00662-f003:**
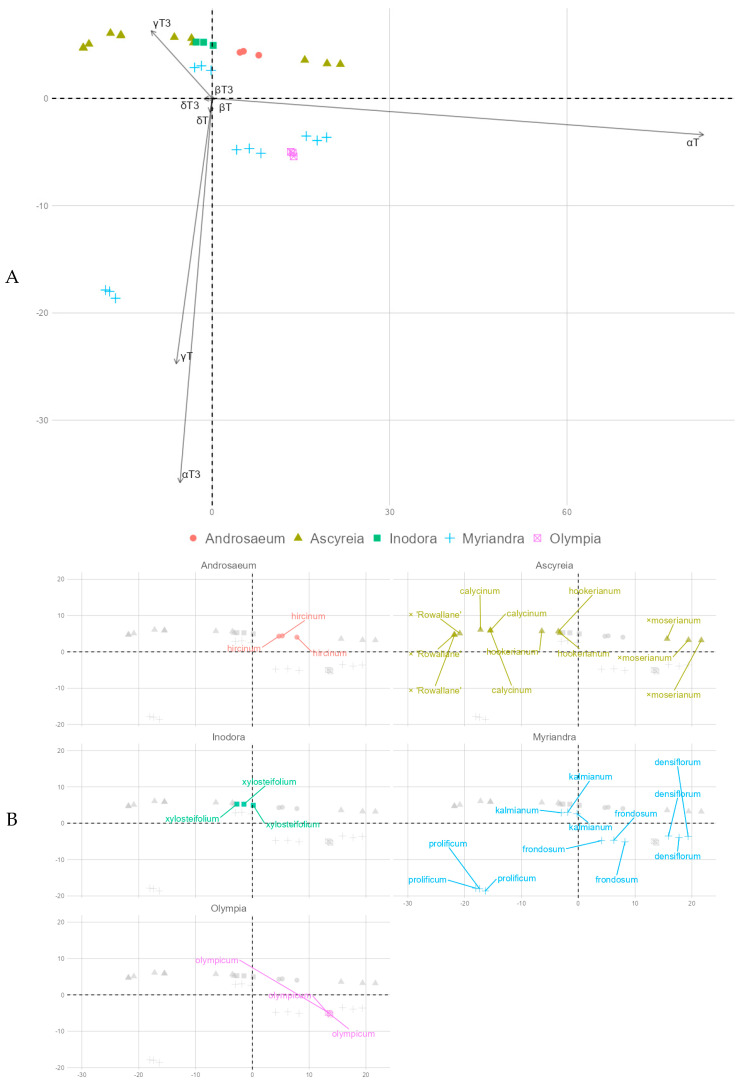
PCA principal component loadings. (**A**)—PC 1 and PC 2 biplot; (**B**)—individual datapoint coordinate plots, separated and highlighted by section, with species denotation. Greyed-out datapoints represent the rest of the dataset. Overlapping labels in contains tocochromanol variables with insignificant principal component loadings (β-tocopherol and tocotrienol and δ-tocopherol and tocotrienol).

## Data Availability

The data used to support the findings of this study are available in the Appendix A and from the corresponding author upon request.

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
