# Peer review of "Tocotrienols in Eleven Species of Hypericum Genus Leaves"

_molecules, 2025, doi:10.3390/molecules30030662_

Round 1
Reviewer 1 Report
Comments and Suggestions for Authors
Main Review:
1.The clarity of Figure 1 and Figure 3 could be improved.
2.Why was 96.2% ethanol chosen to saponify leaf samples? Was this concentration considered in experimental design or previous experience? What impact does it have on the results of the saponification reaction?
3. Why were the two hybrids H. ×moserianum and H × 'Rowallane' selected? What is the significance of analyzing the tocopherol and tocotrienol content of the hybrids?
4. Why does the figure only show the chromatograms of H. calycinum and H. hookerianum? Are there supplementary materials for the images of other species?
5.When comparing with previous studies, the differences in tocochromanol composition and content compared with some studies on Hypericum perforatum were only briefly mentioned, without a comprehensive analysis of the reasons for the differences, including the combined effects of experimental methods (such as extraction solvents, analytical techniques, etc.), plant variety differences, and growth environment differences on the results. A more detailed comparison and discussion should be made to highlight the uniqueness and reliability of this study.
Author Response
We sincerely thank you for the constructive criticism, comments, remarks, and suggestions that have contributed to enhancing the manuscript and its scientific quality. The graphical abstract, manuscript, and supplementary materials have been improved accordingly. Provided changes are marked in red font. For literature we used references manager software therefore changes are not highlighted.
Reviewer 1.
Comments and Suggestions for Authors
Comments 1: [The clarity of Figure 1 and Figure 3 could be improved.]
Response 1: [Thank you for the comment. The figure has been improved. Page 3, page 6-7.]
Comment 2: [Why was 96.2% ethanol chosen to saponify leaf samples? Was this concentration considered in experimental design or previous experience? What impact does it have on the results of the saponification reaction?]
Response 2: [Thank you for pointing this out. We have tested both absolute and 96.2% ethanol for saponification, and did not observe statistically significant differences in extraction efficiency. We have continued using 96.2% for saponification because it is cheaper than absolute ethanol. To be clear we added a short paragraph informing that we did the comparison test and the results are presented in the supplementary materials. Page 9, paragraph 3.3.]
Comment 3: [Why were the two hybrids H. ×moserianum and H × 'Rowallane' selected? What is the significance of analyzing the tocopherol and tocotrienol content of the hybrids?]
Response 3: [Thank you for the comment. The species in the study represent all the possible species which are grown in the National Botanical Garden of Latvia. We investigate all possible Hypericum plants we were able to obtain, including two hybrids, for the most comprehensive data pool possible. We add the note about this fact in to the manuscript. Page 9, top.]
Comment 4: [Why does the figure only show the chromatograms of H. calycinum and H. hookerianum? Are there supplementary materials for the images of other species?]
Response 4: [Thank you for pointing this out. All chromatograms have been added to the Supplementary Materials, including standards of tocopherols and tocotrienols for comparing peak fit to test sample. Slight shifting (left and right) is evident in some chromatograms due to very sharp peaks and system sensitivity to any room environment changes (movement of the staff in the room – small temperature fluctuations in an otherwise temperature-controlled room). The provided chromatograms in the main text and supplementary materials were obtained by different column and chromatographic conditions (testing conditions, Kinetex PFP, 250×4.6 mm, 5 µm) due to technical problems with the PC on which the original chromatograms were obtained two years ago. The chromatogram provided in the main text was obtained two years ago before computer system exchange. The chromatograms provided in supplementary materials were obtained about 18 months ago.]
Comment 5: [Compared with previous studies, and some studies on Hypericum perforatum, the differences in the composition and content of tocopherols were only briefly mentioned, and no comprehensive analysis was conducted on the causes of the differences, including the combined effects of experimental methods (such as extraction solvents, analytical techniques, etc.), plant variety differences, and growth environment differences on the results. A more detailed comparison and discussion should be conducted to highlight the uniqueness and reliability of this study.]
Response 5: [Thank you for the comment. The brief discussion has been supplemented with some more detail, but is limited by the novelty of the material. However, information on tocochromanol content in the genus is very scarce, and H. perforatum, which has been studied the most, is not included in the study, nor are any other species from its section (Hypericaceae). Additional population studies would be needed to asses tocochromanol content and profile variation within a species. Page 5-6.]
Reviewer 2 Report
Comments and Suggestions for Authors
Manuscript ID: molecule-3419029
Manuscript Title: Tocotrienols in eleven species’ of Hypericum genus leaves
This paper presents investigation research about the presence of tocotrienols in eleven Hypericum species’ leaves, since the frequency of tocotrienol derivatives in Clusiaceae species. Authors claimed that any presence of tocotrienols in the Hypericum genus has gone unreported. I think this is an important discovery in the field of nature sciences. The idea is interesting and lots of work had been done for the verification of the mechanism. However, there are still some questions need to be clarified or detailed.
Here are some comments for your references.
#1. The potential applications of this investigated tocotrienols should be briefly analyzed to highlight the importance of this study work.
#2. Since eleven Hypericum species’ leaves contains tocotrienols, the chromatograms of all these eleven experiments should be posted in Figure 1 as original data for comparation.
#3. A brief introduction about the detection approaches (here are HPLC) should be discussed, and interfere peak of the chromatograms should also be explained.
#4. As I know PCA is an algorithm for dimensionality reduction analysis and feature extraction, so what is the purpose of PCA analysis in this research? Or what information can we get from the two diagrams of Figure 3? Since PCA is an abbreviation, its full name needs to be given when it first appears.
Author Response
We sincerely thank you for the constructive criticism, comments, remarks, and suggestions that have contributed to enhancing the manuscript and its scientific quality. The graphical abstract, manuscript, and supplementary materials have been improved accordingly. Provided changes are marked in red font. For literature we used references manager software therefore changes are not highlighted.
Reviewer 2.
Comments and Suggestions for Authors
Manuscript ID: molecule-3419029
Manuscript Title: Tocotrienols in eleven species’ of Hypericum genus leaves
This paper presents investigation research about the presence of tocotrienols in eleven Hypericum species’ leaves, since the frequency of tocotrienol derivatives in Clusiaceae species. Authors claimed that any presence of tocotrienols in the Hypericum genus has gone unreported. I think this is an important discovery in the field of nature sciences. The idea is interesting and lots of work had been done for the verification of the mechanism. However, there are still some questions need to be clarified or detailed. Here are some comments for your references.
Thank you for the positive overview.
Comment 1: [The potential applications of this investigated tocotrienols should be briefly analyzed to highlight the importance of this study work.]
Response 1: [Thank you for pointing this out. The potential application was highlighted in the conclusion part. Page 10.]
Comment 2: [Since eleven Hypericum species’ leaves contains tocotrienols, the chromatograms of all these eleven experiments should be posted in Figure 1 as original data for comparation.]
Response 2: [Thank you for pointing this out. Since it 11 chromatograms, we thinking that placing them in the Supplementary Materials would be best option and this is what we did.]
Comment 3: [A brief introduction about the detection approaches (here are HPLC) should be discussed, and interfere peak of the chromatograms should also be explained.]
Response 3: [Thank you for pointing this out. The detection approaches and the possibility of the interfered peaks of the chromatograms were discussed. Page 4, top.]
Comment 4: [I know PCA is an algorithm for dimensionality reduction analysis and feature extraction, so what is the purpose of PCA analysis in this research? Or what information can we get from the two diagrams of Figure 3? Since PCA is an abbreviation, its full name needs to be given when it first appears.]
Response 4: [Thank you for pointing out. The large amount of variable relative to the sample number makes it difficult to judge which species or sections are similar across the profile. PCA was included to depict similarities between species and main contributors to variation within the genus. The figure was improved as much as we could. Page 6-7.]

Reviewer 3 Report
Comments and Suggestions for Authors
The manuscript "Tocotrienols in eleven species' of Hypericum genus leaves" is a detailed investigation of the lipophilic profile of tocochromanols in leaves, not previously studied, of several Hypericum plant species divided into five sections. The results are clearly presented and provide some relationships between section and tocochromanol composition that may be useful to natural product scientists. Several recommendations could be made to improve the quality and readability of the manuscript.
- The choice of the five sections is not discussed in any detail, in contrast to the large amount of information on the Hypericum species that is not related to the main research question.
- The PCA, which should show the significance of the differences between the sections, is not satisfactorily explained. The quality of Figure 3 is not sufficient. Attempts to use more advanced chemometric methods could possibly provide a better differentiation of the different sections and a better insight into the results.
- The texts in rows 87-108 and 235-246 are practically identical without a clear reason.
- The conclusion does not really provide sufficient information for a complete investigation.
-r. 341 - typing error
Author Response
We sincerely thank you for the constructive criticism, comments, remarks, and suggestions that have contributed to enhancing the manuscript and its scientific quality. The graphical abstract, manuscript, and supplementary materials have been improved accordingly. Provided changes are marked in red font. For literature we used references manager software therefore changes are not highlighted.
Reviewer 3.
Comments and Suggestions for Authors
The manuscript "Tocotrienols in eleven species' of Hypericum genus leaves" is a detailed investigation of the lipophilic profile of tocochromanols in leaves, not previously studied, of several Hypericum plant species divided into five sections. The results are clearly presented and provide some relationships between section and tocochromanol composition that may be useful to natural product scientists. Several recommendations could be made to improve the quality and readability of the manuscript.
Thank you for the positive overview.
Comments 1: [The choice of the five sections is not discussed in any detail, in contrast to the large amount of information on the Hypericum species that is not related to the main research question.]
Response 1: [Thank you for the comment. The sections are provided for context, and did not affect what species were included in the study. They are provided to loosely group the species by their genetic and morphological similarity. The present study investigated all possible Hypericum plants grown in the National Botanic Garden of Latvia in Salaspils, including two hybrids, for the most comprehensive data pool possible. This information was supplemented in method part “Plant material”. Page 9, top.]
Comments 2: [The PCA, which should show the significance of the differences between the sections, is not satisfactorily explained. The quality of Figure 3 is not sufficient. Attempts to use more advanced chemometric methods could possibly provide a better differentiation of the different sections and a better insight into the results.]
Response 2: [Thank you for pointing out. The figure PCA was improved as much as we could. However, additional improvements are limited from our side, due to potential samples which we could choose for the study. As we provided above, we investigated all Hypericum plants grown in the National Botanic Garden of Latvia in Salaspils. Page 6-7.]
Comments 3: [The texts in rows 87-108 and 235-246 are practically identical without a clear reason.]
Response 3: [Thank you for pointing out. Part of this section was removed.]
Comments 4: [The conclusion does not really provide sufficient information for a complete investigation.]
Response 4: [Thank you for pointing out. The conclusion has been improved. Page 10.]
Comments 5: [-r. 341 - typing error]
Response 5: [Our line numbers may be different, the following explanation is for clarity in case the comment refers to “R”. The typing error in “RStudio” has been corrected. In the line before that, “R” refers to the programming language, whose name is the sole letter “R”, similar to other programming languages, of which “C” is the most common (“C++”, a modernized version, is used in MS Office). Like “MatLab” and “python”, “R” is common in data analysis.]

Reviewer 4 Report
Comments and Suggestions for Authors
The manuscript, “Tocotrienols in eleven species’ of leaves of the Hypericum genus” highlights the identification and quantification of tocotrienols, important metabolites for the industry due to their high antioxidant capacity. It is novel and well-written. Some suggestions are described below.
1. The introduction is somewhat extensive, it is suggested to reduce it.
2. Line 153-154 H. perforatum write in italics
3. The authors recently published the article "Tocopherol and tocotrienol homologue recovery from Hypericum perforatum L. and extraction residues after hydroethanolic extraction." It is suggested that it be included in the discussion of the present manuscript.
Author Response
We sincerely thank you for the constructive criticism, comments, remarks, and suggestions that have contributed to enhancing the manuscript and its scientific quality. The graphical abstract, manuscript, and supplementary materials have been improved accordingly. Provided changes are marked in red font. For literature we used references manager software therefore changes are not highlighted.
Reviewer 4.
Comments and Suggestions for Authors
The manuscript, “Tocotrienols in eleven species’ of leaves of the Hypericum genus” highlights the identification and quantification of tocotrienols, important metabolites for the industry due to their high antioxidant capacity. It is novel and well-written. Some suggestions are described below.
Thank you for the positive overview.
Comments 1: [The introduction is somewhat extensive, it is suggested to reduce it.]
Response 1: [Thank you for pointing out. The introduction has been shortened.]
Comments 2: [Line 153-154 H. perforatum write in italics]
Response 2: [Thank you for pointing out. The change has been done. Page 5, top.]
Comments 3: [The authors recently published the article "Tocopherol and tocotrienol homologue recovery from Hypericum perforatum L. and extraction residues after hydroethanolic extraction." It is suggested that it be included in the discussion of the present manuscript.]
Response 3: [Thank you for pointing out. The article has been included in the present paper and discussed. Page 5, second paragraph from middle.]

Round 2
Reviewer 3 Report
Comments and Suggestions for Authors
The present version of the manuscript “Tocotrienols in eleven species’ of Hypericum genus leaves” is much better and can be published with only minor revisions.
The conclusion is not really improved. It is just longer.
There is a small error in the positioning of words in the abbreviation part. Please correct it.
Author Response
The present version of the manuscript “Tocotrienols in eleven species’ of Hypericum genus leaves” is much better and can be published with only minor revisions.
Thank you for the positive overview.
Comments 1: The conclusion is not really improved. It is just longer.
Response 1: Thank you for the comment. The conclusion has been improved.
Comments 2: There is a small error in the positioning of words in the abbreviation part. Please correct it.
Response 2: Thank you for the notification. The abbreviations have been fixed.